# Genetic Characterization of *SWEET* Genes in Coconut Palm

**DOI:** 10.3390/plants14050686

**Published:** 2025-02-23

**Authors:** Jiepeng Chen, Weiming Zeng, Jiali Mao, Runan Chen, Ran Xu, Ying Wang, Ruibo Song, Zifen Lao, Zhuang Yang, Zhihua Mu, Ruohan Li, Hongyan Yin, Yong Xiao, Jie Luo, Wei Xia

**Affiliations:** National Key Laboratory for Tropical Crop Breeding, School of Breeding and Multiplication (Sanya Institute of Breeding and Multiplication)/College of Tropical Agriculture and Forestry, Hainan University, Sanya 572025, China; 22210901000038@hainanu.edu.cn (J.C.); 23210901000057@hainanu.edu.cn (W.Z.); 22210901000081@hainanu.edu.cn (J.M.); 21110901000003@hainanu.edu.cn (R.C.); xuran@hainanu.edu.cn (R.X.); 990811@hainan.edu.com (Y.W.); 20213006581@hainanu.edu.cn (R.S.); 22210901000007@hainanu.edu.cn (Z.L.); yangzhuang@hainanu.edu.cn (Z.Y.); 184457@hainanu.edu.cn (Z.M.); 20223006722@hainanu.edu.cn (R.L.); yinhy@hainanu.edu.cn (H.Y.); jie.luo@hainanu.edu.cn (J.L.)

**Keywords:** *CnSWEET*, gene expansion, evolutionary divergence, promoter activity, sugar transport

## Abstract

Sugar-Will-Eventually-be-Exported Transporters (SWEETs) play a crucial role in sugar transport in plants, mediating both plant development and stress responses. Despite their importance, there has been limited research characterizing the functional characteristics of *CnSWEET* genes in coconut (*Cocos nucifera*). In this study, we conducted a systematic analysis of *SWEET* genes in coconut using bioinformatics, subcellular localization studies, in silico promoter analysis, and functional assays with yeast mutants. A total of 16 *CnSWEET* genes were identified and grouped into four clades. Clade I contained the highest number of genes (eight), derived from four pairs of duplicated genomic segments. In contrast, the other clades had fewer genes (one to four) compared to those in Arabidopsis and other species in the Arecaceae family. An extensive analysis of gene expansion using MSCanX indicated significant divergence in gene expansion patterns, both between and within monocots and dicots, as well as among closely related species within the same family. Notable variations in conserved protein motifs and the number of transmembrane helices (TMHs) were detected within Clade I compared to other clades, affecting the subcellular localization of CnSWEET proteins. Specifically, seven TMHs were associated with proteins located in the cell membrane, while *CnSWEET2A*, which had five TMHs, was found in both the cell membrane and cytosol. Promoter analysis revealed that some *CnSWEET* genes contained endosperm or seed specific motifs associated with specific endosperm expression, consistent with expression patterns observed in transcriptome data. Functional analysis of five *CnSWEET* genes, with transcript sequences supported by transcriptome data, was conducted using yeast mutant complementation assays. This analysis demonstrated diverse transport activities for sucrose, fructose, glucose, galactose, and mannose. Our findings provide valuable insights into the functional diversity of *SWEET* genes in coconuts and their potential roles in stress responses and plant development.

## 1. Introduction

The *SWEET*, short for Sugar-Will-Eventually-be-Exported Transporter, gene family plays a vital role in loading sugars into the phloem for transport throughout a plant, thereby influencing growth, development, and stress responses [1,2,3,4,5,6]. This gene family exhibits considerable structural diversity and can be classified into four clades (I to IV) based on sequence similarity and evolutionary relationships. Members in different clades exhibit differences in subcellular localizations, substrates, and functions [3,4,5,6,7,8]. For instance, in Arabidopsis [9], *AtSWEET8* (*RPG1*) and *AtSWEET13* (*RPG2*) are critical for pollen wall patterning and primexine deposition during the reproductive stage [10,11]. *AtSWEET9* is specifically involved in nectar secretion, facilitating sucrose production that attracts pollinators and enhances reproductive success [12,13]. Additionally, osmotic stress enhances long-distance sucrose transport, a process facilitated by the phosphorylation of *AtSWEET11* and *AtSWEET12*, leading to increased oligomerization [14]. *AtSWEET16* and *AtSWEET17* function as fructose uniporters that regulate fructose levels in leaves and roots [15]. Furthermore, recent research has demonstrated that *GmSWEET6* in soybean is induced by arbuscular mycorrhizae and plays a critical role in mediating sucrose export to AM fungi, which is essential for effective AM symbiosis and influences plant growth and nutrient content [8]. Despite the critical roles of *SWEET* genes in various biological processes, research on these genes in coconuts is currently limited. A systematic analysis of *SWEET* genes in coconut could provide valuable insights into their regulatory functions in coconut fruit development and related processes.

*SWEETs* are integral to the distribution of photosynthates produced during photosynthesis, ensuring that different tissues in a plant can effectively utilize these carbohydrates during various developmental stages, including seed development and fruit ripening. *AtSWEET11*, *12*, and *15* are essential for nutrient transfer from the seed coat to the developing embryo in Arabidopsis thaliana; their absence in a triple mutant leads to significant seed defects [16]. In peach, *PpSWEET9a* and *PpSWEET14* function as sucrose efflux proteins that form a heterooligomer, playing a crucial role in transporting sucrose from source leaves to the fruit [7]. Similarly, research on apple has identified *MdSWEET12a*, a plasma membrane-localized sugar transporter, as a key regulator of sugar unloading in apple fruit [9]. In rice, gene expression patterns and knockout mutants have indicated that *OsSWEET11*, *14*, and *15* are crucial for sucrose transfer during the early stages of rice caryopsis development [17,18,19]. Likewise, in soybean, *GmSWEET15a* and *GmSWEET15b* are essential sugar transporter genes that facilitate sucrose transport from the endosperm to the embryo during early seed development [20]. In maize, *ZmSWEET4c* and its rice ortholog, *OsSWEET4*, are implicated in the transport of hexoses across the basal endosperm transfer layer, directly influencing seed size and weight [21]. *SWEET* genes play a crucial role in fruit development and exhibit tissue-specific expression patterns, making it essential to identify and explore these genes, particularly in the context of coconut fruit maturation, which is a significant developmental process attracting attention.

Plants are often subjected to both biotic and abiotic stresses, and *SWEET* genes have been recognized for their significant role in enhancing plant resilience under these conditions by regulating the transport and distribution of carbohydrates. The expression of *SWEET* genes is regulated by various internal and external factors, including hormone signaling, environmental stimuli, and developmental stages. For instance, research has shown that *SAG29*/*AtSWEET15* are upregulated in response to osmotic stress, facilitating the accumulation of sugars that serve as osmoprotectants [22]. Moreover, the expression patterns of *SWEET* genes can be tissue-specific and stage-dependent, reflecting the physiological demands of the plant [5,9,23,24]. This spatial and temporal specificity of expression regulation is closely linked to the physiological demands of the plant. For example, studies have indicated that two domains within the *AtSWEET11* coding sequence and its promoter are responsible for its specific expression in phloem parenchyma cells [25]. Additionally, *AtSWEET11*/*12* can be regulated by the transcription factor *HOMEOBOX PROTEIN 24*, which influences root growth inhibition in response to salt treatment [26]. In rice, the sugar transporter OsSWEET1b promotes leaf senescence by facilitating the transport of glucose and galactose. Its expression is negatively regulated by the senescence-activated transcription factor *OsWRKY53*, resulting in reduced sugar accumulation and accelerated senescence under both natural and dark-induced conditions [6]. Furthermore, *SWEET* genes are intricately involved in the hormonal signaling pathways that govern plant development and stress responses. For instance, the defect in anther dehiscence in a double *sweet13*/*sweet14* mutant can be reversed by exogenous gibberellic acid (GA) treatment and exhibits altered long-distance transport and responses to GA during germination and seedling stages [27]. Additionally, *LoSWEET14* is activated by *LoABF2* to engage in the abscisic acid (ABA) signaling pathway, promoting the accumulation of soluble sugars in response to various abiotic stresses [28]. The expression of *SWEET* genes in response to hormonal signals further emphasizes their role in coordinating plant responses to environmental changes.

Coconut (*Cocos nucifera*, 2n = 32), a member of the Arecaceae family, is an essential crop and landscape plant primarily found in tropical regions. Known for its versatility, the coconut palm is often referred to as the “tree of life” due to its wide range of uses, which include food, beverages, cosmetics, and construction materials. Recent studies have produced high-quality genomes for both dwarf and tall coconut varieties, allowing for the genetic analysis of key agricultural traits [29,30]. The completion of genome sequencing for various species has facilitated the identification and characterization of many genes. Despite the significant relationships reported between *SWEET* genes and fruit development in many species, research on *SWEET* genes in coconuts remains limited. A systematic analysis of the coconut *SWEET* gene family is essential for identifying and understanding the *SWEET* genes that regulate coconut fruit development, providing a foundation for future studies aimed at enhancing coconut cultivation and fruit quality. In this study, we identified *SWEET* genes within the coconut genome and classified them into four clades based on comparisons with *AtSWEETs*. We analyzed the gene structures, gene expansion, and evolutionary characteristics of *CnSWEETs* in conjunction with data from 14 representative species with available genome sequences. Additionally, we examined the promoter motifs and expression profiles of *CnSWEETs* using transcriptome datasets. Complementary assays with yeast mutants provided insights into the functional divergence of *CnSWEETs*. This research establishes a foundation for further gene function exploration and will contribute to a theoretical framework for analyzing the roles of *CnSWEET*s in coconut seed development.

## 2. Materials and Methods

### 2.1. Identification of SWEET Genes in Coconut and Other Species

The coconut (Cnu) genome sequence, protein sequences, and the transcriptome datasets used in this study were generated from our previous research [29]. Additionally, genome sequences, gene model information, and gene protein sequences for 14 plant species—*Amborella trichopoda* (Atr), *Daucus carota* (Dca), *Solanum tuberosum* (Stu), *Vitis vinifera* (Vvi), *Malus domestica* (Mdo), *Citrus sinensis* (Csi), *Arabidopsis thaliana* (Ath), *Dioscorea alata* (Dal), *Phoenix dactylifera* (Pda), *Elaeis guineensis* (Egu), *Musa acuminata* (Mac), *Ananas comosus* (Aco), *Brachypodium distachyon* (Bdi), and *Oryza sativa* (Osa)—were downloaded from the Phytozome website (https://phytozome-next.jgi.doe.gov/) and the National Center for Biotechnology Information (NCBI) website (https://www.ncbi.nlm.nih.gov/).

The hidden Markov model (HMM) profiles of the SWEETs’ conserved domains (MtN, PF03083) were downloaded from the Pfam database (http://pfam.xfam.org/) and used to search for the SWEET proteins in the coconut proteome with HMMER software (v3.4). In addition, the protein sequences of 17 AtSWEETs were downloaded from TAIR (https://www.arabidopsis.org/) and used as queries to search in the coconut proteome. All resulting non-redundant protein sequences were analyzed by the MEME online software (https://meme-suite.org/meme/tools/meme, accessed on 10 October 2024). The distributions of transmembrane helices (THMs) were predicted using the TMHMM online software (v2.0, http://www.cbs.dtu.dk/services/TMHMM, accessed on 26 October 2024).

### 2.2. SWEET Gene Structure, Phylogenetic Analysis, and Duplication Event Identification

The gene structures of *CnSWEETs* were manually corrected, based on the transcriptome dataset, and visualized using TBtools (v2.154) [31]. The duplicated genomic segments for each species were displayed in circle by TBtools [31]. Multiple sequence alignment was carried out for *CnSWEETs* and *AtSWEETs* and used to construct a neighbor-joining (NJ) tree, which was constructed by MEGA 7.0 with 1000 bootstrap replicates [32]. The duplicated *SWEET* genes for coconut and 14 other species were identified by MCScanX (v2) according to the previously described criteria [33].

### 2.3. Gene Expression Pattern Analysis Based on Transcriptome Datasets

The RNA-seq SRAs (CRA004778, https://ngdc.cncb.ac.cn/gsa, accessed on 10 October 2023) created in our previous research were used in this study. The transcriptome dataset includes RNA-seq data for five types of tissues: leaf, flower, stem, endosperm, and mesocarp. The FPKM values were calculated as described in our previous research, using Hisat2 (v2.2.0) for read mapping and Stringtie (v2.2.1) for isoform assembly [33,34]. The expression correlation between duplicated *CnSWEET* genes was measured using the Pearson correlation coefficient (P.C.C.). The significance of the P.C.C. value was tested by the *t* test (cor.test in R).

### 2.4. Promoter Analysis

We extracted the upstream 2000 bp sequences starting from the start codon of each *CnSWEET* gene. The analysis of promoter motifs was conducted using the online websites Plantcare (http://bioinformatics.psb.ugent.be/webtools/plantcare/html/, accessed on 10 September 2024) and TSSP (http://www.softberry.com/, 11 September 2024) [35]. The putative TATA-box, CAAT-box, and motifs associated with tissue specific expression and phytohormone responsiveness were identified using TBtools.

### 2.5. Transient Expression of CnSWEETs in Tobacco Epidermal Cells for Subcellular Localization

Total RNA and cDNA samples were derived from mixed tissues containing leaves, flowers, endosperm, and mesocarp tissues, as described in our previous research [33]. The full-length coding sequences of *CnSWEETs* were amplified using the primers listed in Appendix A. The same methods described above were applied to construct the *CnSWEET* OE-expression vectors, specifically the pc1300-35S-CnSWEET-eGFP vector.

Agrobacterial cultures of 35S::CnSWEET:eGFP and 35S::OsCBLn1:RFP (as positive controls for cell membrane localization) fusion constructs were pelleted and resuspended in the infiltration media. Empty vectors, pc1300-35S-eGFP and pc1300-35S-RFP, were used as negative controls. The same transformation protocol was used as above, and GFP signals were detected at time intervals of 48–72 h post-infiltration using a confocal microscope (SESIS, LMS980, Zeiss, Oberkochen, Germany).

### 2.6. Substrate Specificity Analysis of CnSWEET Proteins in Yeast

The coding sequences (CDSs) of *CnSWEETs* were amplified using specific primers and subsequently sub-cloned into the yeast expression vector pDR196. The primers used are listed in Appendix A. The resulting constructs, along with the empty pDR196 vector as a negative control, were transformed into the yeast mutants EBY.VW4000 and SUSY7/ura3 for sugar uptake assays. For the EBY.VW4000 cells, serial dilutions (1:10, 1:100, and 1:1000) were plated on solid SD media supplemented with 1% maltose (as the control) or with fructose, glucose, galactose, and mannose. In the case of the SUSY7/ura3 strain, cells were also serially diluted 10-fold (1:10, 1:100, and 1:1000) and spotted onto solid SD media containing either 2% glucose (as the control) or 2% sucrose. The plates containing the transformants were photographed after incubation at 28 °C for 3 to 5 days.

## 3. Results

### 3.1. SWEET Gene Family in Coconut Genome

To identify SWEET genes in coconut, we conducted a hidden Markov model (HMMER) search against the coconut genome protein database using the MtN3_slv domain (PF03083) as a seed. The conserved MtN3 domains were detected by verifying sequences in the Pfam databases. Sixteen CnSWEET proteins with complete domains were obtained and designated according to the best hits of *AtSWEET* homologs (Appendix A). The *CnSWEET* genes could be classified into four clades. The naming and clade analysis of *CnSWEET*s were conducted through comparative analysis with Arabidopsis genes. Coconut had the highest number of *SWEET* genes in Clade I (eight), followed by Clade III (four), Clade II (three), and Clade IV (one) (Figure 1A). The gene structures of *CnSWEETs* were validated using previously published transcriptomic and full-length transcript data for coconut [30]. In Clade I, the gene structures of *CnSWEET1A*, *CnSWEET2A*, *CnSWEET2C*, and *CnSWEET3B* were supported by transcript data, which included the untranslated region sequences (UTRs). *CnSWEET1B* and *CnSWEET3A* had gene regions of approximately 24 kb and 11 kb, respectively, due to the insertion of transposons in their intronic regions. Seven genes in this family contained six exons, while *CnSWEET2B* had eight exons. Clade II included three gene members, with *CnSWEET4* and *CnSWEET5* exhibiting six exons, while *CnSWEET6* had five exons. Clade III comprised four gene members, all of which had six exons, and *CnSWEET15* included information about UTR sequences. Clade IV consisted of only one gene, which also had six exons.

Analysis of the conserved motifs in CnSWEET proteins revealed the presence of seven significant conserved sequences identified by the MEME online software (Figure 1B). Genes in Clade I exhibited the most divergent protein domain variations, with the number of conserved domains ranging from four (in CnSWEET1B and CnSWEET3B) to seven (in CnSWEET1A, CnSWEET3A, and CnSWEET11/13/15). The absence of certain conserved domains in CnSWEET1B may be attributed to transposon insertion. In contrast, genes in the remaining clades displayed six highly conserved motifs, albeit with variations in the C-terminal domain.

The TMHMM online software analysis suggested the presence of transmembrane regions and the orientation of CnSWEETs (Figure 1B). The distribution of transmembrane helices (TMHs) largely overlapped with the six conserved domains detected by the above but did not correspond precisely to specific regions, nor did it account for the domains in the N-terminus or the peptides in those areas. Notable variations in the number of TMHs were observed in Clade I and Clade IV, with counts ranging from three (*CnSWEET2D*) to six (*CnSWEET2B*, *CnSWEET3A*, and *CnSWEET16*). The absent TMH in *CnSWEET3A* may have resulted from transposable element insertion in the intron. Additionally, there were seven conserved TMHs for most *CnSWEETs*, which aligns with findings in other reported species [36,37].

### 3.2. The Evolutionary Feature of the SWEET Gene Family

The *CnSWEETs* could be categorized into four clades when grouped with the *AtSWEET* members, and there was a notable difference in the trend of member increase between the two species (Figure 2A). Coconut exhibited the highest number of members in Clade I, while Clade I contained only three *AtSWEET* members, each associated with two to four *CnSWEET* members. In contrast, there were fewer *CnSWEE*T members in the other three clades compared to Arabidopsis. Clade II included three *CnSWEETs* compared to five *AtSWEETs*, while Clades III and IV had a greater number of *AtSWEETs* than *CnSWEETs*. This discrepancy may be attributed to the tetraplication of *AtSWEETs* in Clade III (*AtSWEET10*/*11*/*12*/*14*), while Clade II (*AtSWEET4*/*8*) and Clade IV (*AtSWEET16*/*17*) experienced only one duplication event (Figure 2B).

The expansion of *CnSWEETs* occurred through segmental duplication (Figure 2C). In Clade I, there were four duplicated pairs: *CnSWEET1A*/*1B*, *CnSWEET2A*/*2C*, *CnSWEET2B*/*2D*, and *CnSWEET3A*/*3B*. Additionally, there was one duplicated pair in Clade II (*CnSWEET4*/*5*) and one in Clade III (*CnSWEET11*/*15*). When examining homologous segments containing *CnSWEETs* or *AtSWEETs* between the two species, a total of thirteen pairs were identified (Figure 2D). In Clade I, three pairs were found: *CnSWEET2C*/*AtSWEET2*, *CnSWEET3A*/*AtSWEET3*, and *CnSWEET3B*/*AtSWEET3*. This indicates that the duplication events of *CnSWEET3A* and *CnSWEET3B* shared the same synteny as *AtSWEET3*. In Clade II, only one homologous segment was detected: *CnSWEET4*/*AtSWEET4*. In Clade III, two *CnSWEETs* (*CnSWEET11*/*15*) were found in homologous segments with each of the three *AtSWEETs* (*AtSWEET11*/*13*/*15*), which resulted from segmental duplication in both species. Finally, in Clade IV, *CnSWEET16* was located in a syntenic segment with that of the duplicated pair *AtSWEET16*/*17*.

The comparison of *SWEET* genes across 15 species, including the basal extant flowering plant *Amborella trichopoda* (Atr), six dicot species, and eight monocot species, revealed a widespread divergence in *SWEET* gene expansions among these species (Figure 3). Using the same methodology applied to coconut, we identified the *SWEET* gene members from coconut and the other fourteen plant species, with their gene ID and subclade information detailed in Appendix A. *Amborella trichopoda* (Atr) was considered the ancestral status for both monocot and dicot species, with the fewest *SWEET* genes: nine, with just one duplication in Clade III. Therefore, the ancestral copy number state for Clades I to IV was represented by four, two, one, and one gene members, respectively.

When analyzing the genomic duplication events in all six dicots, we can explain some of the gene expansions. Among the six dicots, *Daucus carota* (Dca), *Solanum tuberosum* (Stu), and *Malus domestica* (Mdo) exhibited two, one, and five instances of segmental duplication, respectively, resulting in a greater number of gene members compared to *Vitis vinifera* (Vvi), *Citrus sinensis* (Csi), and *Arabidopsis thaliana* (Ath). These three species had no duplication events and possessed three to four gene members, similarly to Atr. Notably, Ath may have lost one *SWEET* gene. In Clade II, an increase in gene members ranging from four to seven was observed. While Stu, Mdo, and Csi showed detectable duplication or triplication events contributing to gene expansion, such events were not detected in Ath and Dca. All species in Clade III exhibited an increase in gene numbers, with Dca (15), Stu (15), and Mdo (9) having the highest counts of *SWEET* genes, followed by Ath (7), Vvi (5), and Csi (5). For Dca, two duplication events and one tetraplication contributed to gene expansion, while only one duplication event was detected in both Stu and Mdo. In Arabidopsis, one tetraplication was identified in Clade III. In contrast, Clade IV exhibited only one to three gene members across all six species, with fewer duplication events detected.

On the other hand, divergent gene expansion occurred among the eight monocots, both within these species and in comparison to the dicots analyzed earlier. Most species in Clade I had more than four gene members, with one, two, or even four instances of duplication detected, resulting in six to eight gene members. *Musa acuminata* (Mac) and *Ananas comosus* (Aco) had five and four gene members, respectively, with no detectable segmental duplication in this clade. Coconut exhibited the most notable expansion, with four duplications in Clade I, compared to the two duplications observed in *Elaeis guineensis* (Egu) and one duplication in *Phoenix dactylifera* (Pda), both closely related species in the Arecaceae family. Nevertheless, the number of gene members in these species remained similar, ranging from six to eight. In Clade II, coconut had the fewest gene members, with only three, which was just one more than those in Atr (two), resulting from one duplication event in coconut. These were significantly fewer than the gene members found in Egu and Pda. In comparison, the remaining monocots had gene members ranging from four (*Oryza sativa*, Osa) to eight (*Brachypodium distachyon*, Bdi), despite belonging to the same Poaceae family. Both triplication and duplication events contributed to gene expansion in these species.

Additionally, extensive variation in gene numbers was observed in Clade III, ranging from two (Pda) to eleven (Mac). Mca had the highest count of eight gene members, supported by two duplication events and one tetraplication event. *Dioscorea alata* (Dal, eight) and Aco (six) each had one triplication event, followed by Osa which had one duplication event as well. Similarly to the dicots, limited gene expansion was detected in Clade IV, with only one duplication occurring in both Mac and Aco.

### 3.3. The Promoter Character and Expression Profile of CnSWEET Genes

We observed significant variation in expression patterns for the sixteen *CnSWEET* genes, both within and between clades. Promoter motif analysis showed extensive divergence (Figure 4A). According to TSSP analysis, only *CnSWEET2B* lacked a putative promoter motif. Additionally, analysis using the PlantCARE website identified the conserved promoter motif CAAT box in most of the promoters, albeit at varying distances from the start codon. Tissue-specific expression motif analysis indicated that *CnSWEET2B*, *CnSWEET3B*, *CnSWEET9*, and *CnSWEET11* contained motifs associated with meristem expression. For the endosperm expression motif GCN4, identified in rice [38], six gene promoters contained this motif, while seed-specific motifs were detected in *CnSWEET5* and *CnSWEET7*. Expression patterns derived from twelve sets of coconut tissues demonstrated tissue-specific expression for most genes (Figure 4B). Notably, *CnSWEET15* exhibited both seed-specific and endosperm-specific motifs, showing high FPKM values, specifically in the endosperm. Furthermore, genes in Clade II and *CnSWEET9* in Clade III displayed significantly higher expression in male flowers. *CnSWEET2A* and *CnSWEET2C* showed relatively high expression in leaves and shared a significant co-expression pattern, as they were located in homologous segments. In contrast, there were low-expression correlations between other pairs of duplicated genes. For instance, *CnSWEET2A* exhibited relatively high expression in mesocarp, while *CnSWEET2C* showed slightly higher expression in male flowers (Figure 4C).

### 3.4. Subcellular Location of CnSWEET Proteins

SWEET proteins were identified as possessing transmembrane helices, which are believed to play a functional role when embedded in membranes. According to the transcriptome analysis and ISO-seq dataset used for gene model correction, five *CnSWEET* genes—*CnSWEET1A*, *CnSWEET2A*/*2B*, *CnSWEET15*, and *CnSWEET6*—were supported by ISO-seq transcript sequences (Figure 1A). For subcellular analysis, the above five *CnSWEET* genes were selected based on reliable transcript sequences. The subcellular localizations of CnSWEET proteins were determined by transiently expressing CnSWEET-GFP fusion proteins in tobacco epidermal cells. The GFP signals were compared with those of a previously reported membrane-localized protein, OsCBL1n-RFP. Fluorescence signals for four CnSWEET-GFP proteins, except for CnSWEET2A-GFP, were specifically observed at the cell membrane. In contrast, CnSWEET2A-GFP was localized to both the cell membrane and the cytosol, appearing in a vesicle-like form (Figure 5A). Among the five proteins, CnSWEET1A, CnSWEET2C, and CnSWEET15 exhibited seven conserved transmembrane helices, whereas CnSWEET2A and CnSWEET6 displayed variations in the number of transmembrane helices (Figure 1B). Three-dimensional structure predictions using SWISS-MODEL confirmed the transmembrane predictions, indicating that CnSWEET15 and CnSWEET16 possessed a long stretch of soluble C-terminal ends (Figure 5B). The localization results also indicated that CnSWEETs with seven transmembrane helices showed a conserved subcellular localization, whereas CnSWEET2A, with five transmembrane helices, exhibited variability.

### 3.5. CnSWEET Proteins Exhibit Variable Tendencies in Sugar Transport

Two yeast mutants, EBY.VW4000 and SUSY7/ura3, were used to test the sugar transport ability for the above five *CnSWEETs*, which belonged to Clade I, III, and IV. The yeast expression vectors—pDR196 inserted with *CnSWEET1A*, *CnSWEET2A*/*C*, *CnSWEET15*, or *CnSWEET16*—were transformed into EBY.VW4000 and SUSY7/ura3. Compared to the transformants expressing pDR196, CnSWEET1A was able to uptake glucose effectively, followed by the uptake of galactose (Figure 6A). *CnSWEET2A* facilitated the uptake of fructose, galactose, and mannose but was not as effective for glucose, while *CnSWEET2C* was able to restore the growth on the medium supplemented with galactose and mannose (Figure 6A). *CnSWEET15* and *CnSWEET16* enabled glucose and mannose uptake in the yeast EBY.VW4000 mutant (Figure 6A). In addition, *CnSWEET2A*, *CnSWEET15*, and *CnSWEET16* conferred sucrose uptake in the yeast SUSY7/ura3 mutant (Figure 6B).

## 4. Discussion

This study provides a comprehensive analysis of the structural differences and similarities among the *CnSWEET* gene subfamilies in coconut, with a particular emphasis on gene structure and protein domains. The findings reveal notable disparities in the expansion patterns of subfamily members between coconut and fourteen other angiosperm species, highlighting the evolutionary dynamics of the *SWEET* gene family. Furthermore, the investigation uncovers significant heterogeneity in the expression of various *CnSWEET* genes, suggesting a correlation with their functional differentiation. These insights establish a foundational understanding for future research into the specific functions of *CnSWEET* genes and their overall roles in the growth and development of coconut.

The *SWEET* gene family plays a crucial role in sugar transport and metabolism across various plant species. The amplification of *SWEET* genes is a common phenomenon in monocots and dicots, and we detected distinct variations in the expansion of *SWEET* genes, which could contribute to functional diversification (Figure 3). Similarly, research across three peanut species has revealed variations in the number of *SWEET* gene members, with 47 in *Arachis hypogea*, 23 in *Arachis duranensis*, and 24 in *Arachis ipaensis*. Analysis of 11 species indicated that whole-genome duplication, segment duplication, and tandem duplication were the primary mechanisms contributing to the expansion of SWEET genes [3]. Gene amplification frequently results in sub-functionalization, wherein duplicated genes undergo evolutionary changes to take on specialized roles in specific developmental contexts. The coconut plant exhibited a significant expansion in Clade I, which also contained the most divergent CnSWEET proteins. Many studies have demonstrated that certain *SWEET* genes may be preferentially expressed in roots, leaves, or flowers, allowing plants to finely tune their sugar transport mechanisms according to developmental needs and environmental conditions [19,27,36,39]. *CnSWEETs* showed high divergence in expression for duplicated gene pairs (Figure 4), as well as transport variations in sugar uptake (Figure 6). As a result, the functional divergence of *CnSWEET*s could be deduced and can provide clues for future research.

*CnSWEETs* have divergent protein structures within clades and between clades. This structural variation, particularly in the number of transmembrane helices (TMHs), suggests that it could significantly influence the subcellular localization of these proteins. For instance, the presence of differing TMH counts might dictate whether CnSWEET proteins are predominantly situated in the plasma membrane or found in intracellular compartments. SWEET proteins are primarily localized to the plasma membrane, where they play a crucial role in facilitating the export of sugars from source tissues, such as leaves, to sink tissues, like roots, fruits, and seeds [2]. This transport is vital for plant growth and development, ensuring that energy-rich compounds are efficiently distributed where they are needed most. Additionally, the discovery that some SWEET proteins are localized in intracellular compartments, including the endoplasmic reticulum, Golgi membranes, and the tonoplast, suggests that these proteins may also be involved in sugar processing or intracellular transport mechanisms [40]. For instance, *AtSWEET9* functions specifically in nectar secretion, facilitating the production of sucrose for nectar, which attracts pollinators and enhances reproductive success [12,13]. The heterogeneity of different SWEET proteins in coconuts may determine their functional site variations.

Furthermore, the spatiotemporal expression of *SWEET* genes exhibits significant heterogeneity among different members, closely linked to their functional differentiation [2,5,13,24,41]. Specific promoter elements have been identified as key regulators of this expression variability, determining when and where each gene is activated or repressed. For instance, two domains within the coding sequence and promoter of *AtSWEET11* specifically regulate its expression in phloem parenchyma cells [25]. In *CnSWEETs*, noticeable variations in the promoter region include different tissue-specific expression motifs, which likely contribute to their unique spatiotemporal expression patterns (Figure 4). Similar specific expression controls have been observed in other species, including Arabidopsis, pineapple, and lichi [13,24,25,41,42]. Transposon insertion in the intron region also appear to drive the protein variations within CnSWEETs. Similar results have been reported for the lower gene expression level caused by transposon insertion [43]. Additionally, an increasing number of transcription factors, such as *HOMEOBOX PROTEIN 24*, *OsWRKY53*, and *LoABF2*, have been identified as regulators of *SWEET* genes [6,26,28]. This variation in promoter motifs may also lead to distinct regulatory modes. Collectively, these findings indicate that different CnSWEET proteins possess unique functions and are active in specific tissues.

SWEETs exhibit diverse substrate affinities, which may be attributed to their unique structural features and the evolutionary pressures exerted by varying environmental conditions [8,9,14,44,45]. The investigation in the study indicates that *CnSWEET1A* is highly efficient in transporting glucose, making it essential for energy supply, particularly in tissues with high metabolic activity. In rice, OsSWEET1b also can import glucose and galactose [6]. Conversely, *CnSWEET2A*’s ability to preferentially transport fructose and galactose indicates a specialized role that may be vital in certain developmental contexts or stress responses, where these sugars are more readily available or necessary for growth. These results are consistent with previous reports that SWEETs in Clade I primarily transport hexoses [46]. The ability of certain SWEETs, such as *CnSWEET15* and *CnSWEET16*, to transport sucrose and glucose suggests that they are well-adapted to facilitate energy distribution during key physiological processes, such as fruit development and stress acclimation. The results demonstrate distinct sugar transport profiles among the tested *CnSWEETs*, highlighting their potential roles in sugar metabolism and transport in plants.

## 5. Conclusions

Coconut has 16 *CnSWEET* genes that exhibit significant structural and functional diversity, along with variations in conserved protein motifs and transmembrane helices. Notably, there is gene expansion in Clade I in coconut, which differs from what is observed in other species. Additionally, there is an evolutionary divergence between monocots and dicots. The presence of endosperm- and/or seed- specific motifs in the promoters of *CnSWEET11* and *15* aligns with their expression patterns in endosperm, further emphasizing their potential roles in plant development. Functional assays confirm that these genes exhibit diverse transport activities for various sugars. Overall, our findings enhance the understanding of *SWEET* gene functionality in coconuts and pave the way for further research into their contributions to plant physiology and adaptation. Future studies could explore the regulatory networks governing *CnSWEET* gene expression and their interactions with other metabolic pathways, providing deeper insights into the role of these genes in the overall growth and resilience of coconut plants.

## Figures and Tables

**Figure 1 plants-14-00686-f001:**
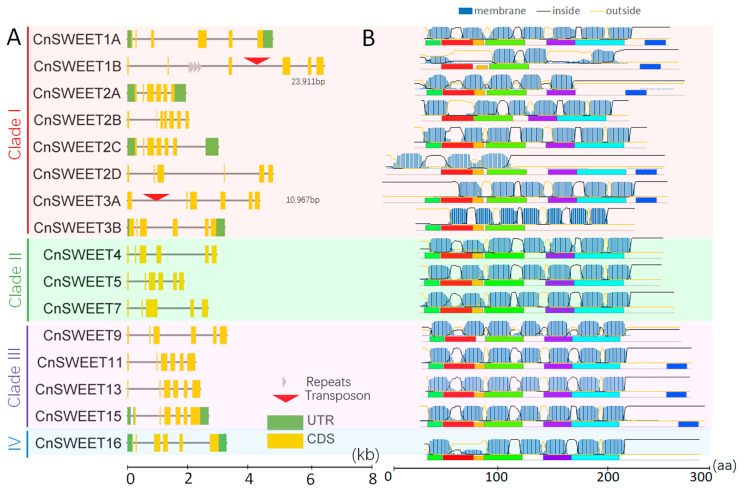
The gene structures and protein features of *CnSWEETs*. (**A**) The gene structures of *CnSWEETs* displayed according to gene model information and ISOseq data modification. The gene structures were displayed using TBtools. Transposon and repeat sequences were analyzed by RepeatMasker (v4.1.1). (**B**) The distribution of conserved protein motifs and transmembrane regions. The former were deduced from the MEME online software. The TMHMM online software was used to analyze the transmembrane regions.

**Figure 2 plants-14-00686-f002:**
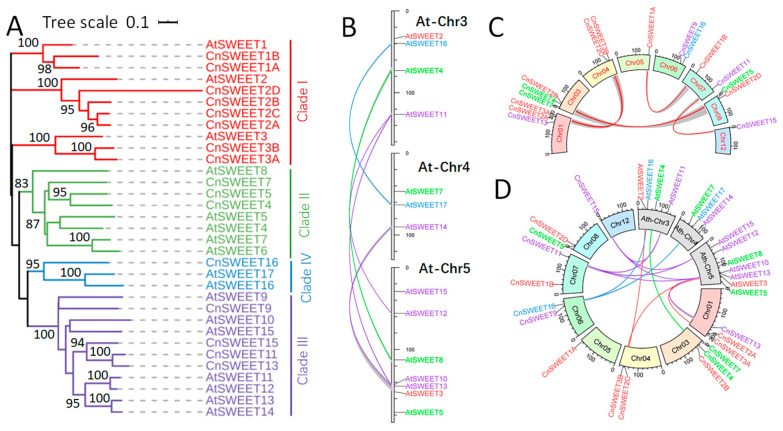
*CnSWEET* gene members in coconut palm and compared with *AtSWEETs*. (**A**) The phylogenetic tree of *CnSWEETs* and *AtSWEETs* was constructed using the neighbor-joining method in MEGA 7.0. (**B**) The genomic locations of *AtSWEET* genes situated within the duplicated genomic segments. (**C**) The genomic locations of *CnSWEET* genes situated within the duplicated genomic segments. (**D**) Homologous genomic segments between coconut and Arabidopsis contain *CnSWEET* and *AtSWEET* genes, respectively. The genomic locations of *CnSWEETs* and *AtSWEETs*, along with duplicated genomic segments containing *SWEETs* deduced from MCScanX analysis, were visualized by TBtools.

**Figure 3 plants-14-00686-f003:**
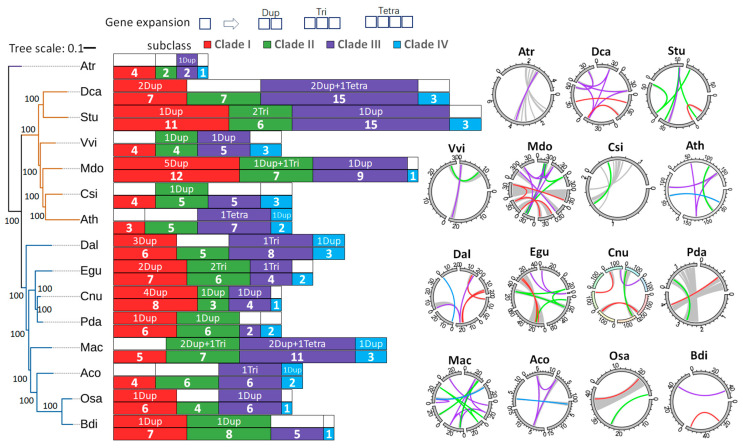
The *SWEET* gene expansion of coconut and fourteen other species in the context of angiosperms. The ML phylogenetic tree of these 15 species was constructed using 140 single-copy genes detected by orthfinder, which was the same method used in our previous research [33]. The method used in this study determined the number, subclade, and duplicated events of *SWEETs* in each species, following the approach used for coconut and Arabidopsis. The length of the colored bar with numbers represents the number of *SWEETs* detected in each clade. The genomic segment duplication events were identified by MCScanX analysis. “Dup”, “Tri”, and “Tetra” represent that the numbers of homologous segments in each duplication event were two, three, and four, respectively. The number before these letters represents the frequency of this type of duplication detected. The synteny regions in chromosomes were displayed in circles using TBtools. The length of chromosomes is measured in megabases. The Materials and Methods Section lists the information about the three characters representing the species. Three-letter abbreviations are used to represent each species: *Amborella trichopoda* (Atr), *Daucus carota* (Dca), *Solanum tuberosum* (Stu), *Vitis vinifera* (Vvi), *Malus domestica* (Mdo), *Citrus sinensis* (Csi), *Arabidopsis thaliana* (Ath), *Dioscorea alata* (Dal), *Phoenix dactylifera* (Pda), *Elaeis guineensis* (Egu), *Musa acuminata* (Mac), *Ananas comosus* (Aco), *Brachypodium distachyon* (Bdi), and *Oryza sativa* (Osa).

**Figure 4 plants-14-00686-f004:**
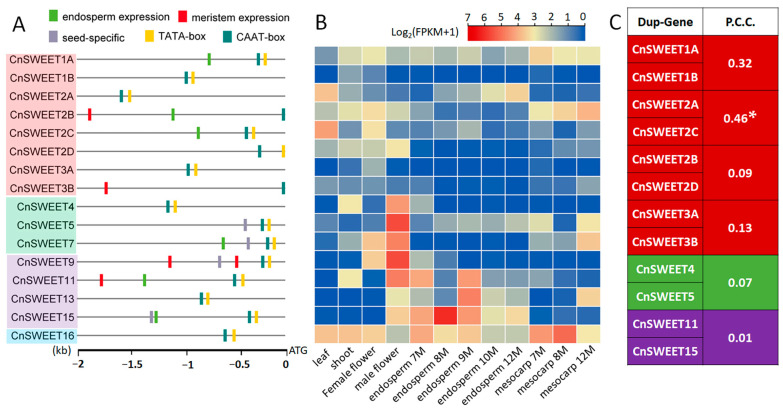
The distribution of the promoter conserved motifs and expression patterns of *CnSWEETs*. (**A**) Motifs in all *CnSWEET* promoters. The motifs in the putative promoters of *CnSWEETs*, located 2000 bp upstream from the ATG start codon, were analyzed using the Plantcare and TSSP software. (**B**) The heatmap of *CnSWEET* expression based on log2-transformed mean FPKM for coconut leaf, shoot, endosperm, and mesocarp tissues was generated using the transcriptomes used in this study. (**C**) The Pearson correlation coefficients of FPKM values for gene pairs were derived from segmental duplication. The Pearson correlation coefficient (P.C.C.) between duplicated *CnSWEET* gene pairs was calculated using the coconut transcriptomes, and significance testing was performed using a *t*-test (* *p* < 0.05) in R (cor.test).

**Figure 5 plants-14-00686-f005:**
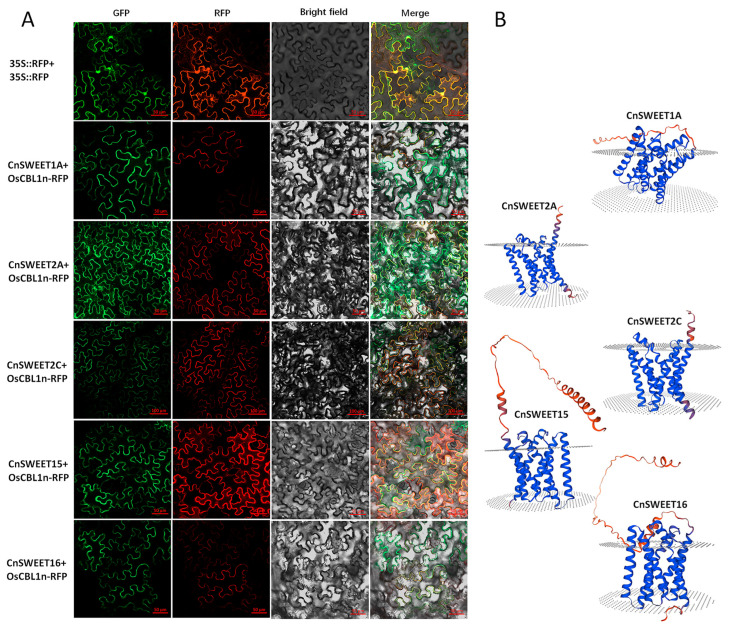
The subcellular location and 3D protein structures of CnSWEET proteins. (**A**) CnSWEET proteins were localized to the membranes. (**B**) The 3D protein structures of CnSWEET protein were deduced and displayed using the online SWISS-MODEL. The 35S::CnSWEET: eGFP fusion protein and the cell membrane marker 35S::OsCBL1:RFP fusion protein were transiently expressed in tobacco epidermal cells. GFP and RFP signals were detected with time intervals between 48 and 72 h post-infiltration using a confocal microscope (SESIS, LMS980). GFP: green fluorescence; RFP: red fluorescence; Bright field: visible light; Merge: visible light merged with fluorescence. Scalebars: 50 μm or 100 μm.

**Figure 6 plants-14-00686-f006:**
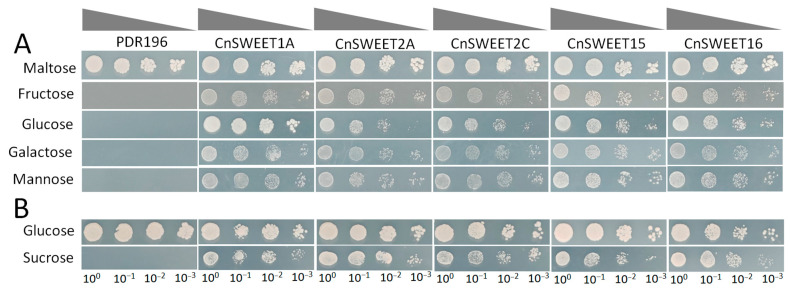
Substrate specificity analysis of five selected CnSWEET proteins in yeast mutant strains EBY.VW4000 (**A**) and SUSY7/ura3 (**B**). Cells were serially diluted 10-fold (10-, 100-, and 1000-fold) and spotted on solid SD media supplemented with 2% concentration of different sugar substrates. Maltose and glucose were only carbon sources used in positive controls for EBY, VW4000 and SUSY7/ura3 cells, respectively.

## Data Availability

Data available in a publicly accessible repository. For transcriptome datasets used in this study were deposited in the website of China National Center for Bioinformation (accession number: CRA004778, https://ngdc.cncb.ac.cn/gsa). For all other genome sequences and gene sequences were downloaded from the Phytozome website (https://phytozome-next.jgi.doe.gov/) and the National Center for Biotechnology Information (NCBI) website (https://www.ncbi.nlm.nih.gov/).

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
