# Peer review of "Genetic Characterization of SWEET Genes in Coconut Palm"

_plants, 2025, doi:10.3390/plants14050686_

Round 1
Reviewer 1 Report
Comments and Suggestions for Authors
Jiepeng Chen et al, did very good effort. Overall, the paper is well written the figures are quite attractive and informative. Please make sure the minor correction, and afterwards endorse the paper for publication.
Abstract
Promoter analysis revealed that some CnSWEET genes contained motifs associated with 29 specific tissue expression, (for example name some of tissue here).
Material method
The gene structure of CnSWEET was manually corrected based on transcriptome dataset and were visualized" – mismatch between singular/plural ("was" vs. "were")
Line 139: "FPKM values were calculated as conducted as our previous research" – revise to calculated as described in our previous research.
Use either "FPKM values" or "expression levels" consistently across the section
Results
Ensure consistent naming for software and methods (e.g., "MEME online" and "TMHMM online" should follow a uniform format).
In "CnSWEET2B has eight exons. Clade II includes three gene members, all exhibiting six to seven exons," the phrase "six to seven exons" could be clarified to specify which genes exhibit the range.
Discussion
Too short, add 1 more paragraph.
Conclusion
The conclusion is missing please write the brief conclusion
Comments on the Quality of English LanguageEnglish is well
Author Response
Reviewer 1
Jiepeng Chen et al, did very good effort. Overall, the paper is well written the figures are quite attractive and informative. Please make sure the minor correction, and afterwards endorse the paper for publication.
Abstract
Promoter analysis revealed that some CnSWEET genes contained motifs associated with 29 specific tissue expression, (for example name some of tissue here).
Material method
The gene structure of CnSWEET was manually corrected based on transcriptome dataset and were visualized" – mismatch between singular/plural ("was" vs. "were")
Line 139: "FPKM values were calculated as conducted as our previous research" – revise to calculated as described in our previous research.
Use either "FPKM values" or "expression levels" consistently across the section
Results
Ensure consistent naming for software and methods (e.g., "MEME online" and "TMHMM online" should follow a uniform format).
In "CnSWEET2B has eight exons. Clade II includes three gene members, all exhibiting six to seven exons," the phrase "six to seven exons" could be clarified to specify which genes exhibit the range.
Discussion
Too short, add 1 more paragraph.
Conclusion
The conclusion is missing please write the brief conclusion
>>>>Response
Thank you for your thoughtful feedback on our manuscript. We appreciate your positive remarks regarding our efforts and the quality of the figures. In response to your comments, we have made the necessary revisions and clarifications as outlined below:
- Abstract: Promoter analysis revealed that some CnSWEET genes contained motifs associated with 29 specific tissue expression, (for example name some of tissue here).
>>>>Response
Thank you for your insightful comment regarding the promoter analysis of CnSWEET genes. We have included specific examples of CnSWEET genes that contain motifs associated with endosperm and seed-specific expression. Notably, these genes show specific expression patterns in the endosperm, which is critical for seed development and nutrient transfer. We appreciate your feedback and have included this information in the revised manuscript to highlight the functional significance of these SWEET genes in tissue-specific expression. This addition enhances the clarity and relevance of our findings.
- Material method
The gene structure of CnSWEET was manually corrected based on transcriptome dataset and were visualized" – mismatch between singular/plural ("was" vs. "were")
>>>>Response
Thank you for pointing out the inconsistency in verb agreement in our statement regarding the gene structure of CnSWEET. We have corrected this error in the revised manuscript to ensure that the verb matches the plural subject. The revised sentence now accurately reflects that the gene structures of CnSWEET were manually corrected based on the transcriptome dataset and visualized. We appreciate your careful review and attention to detail.
Line 139: "FPKM values were calculated as conducted as our previous research" – revise to calculated as described in our previous research.
>>>>Response
Thank you for your suggestion regarding the phrasing in line 139. We have revised the sentence to read: "FPKM values were calculated as described in our previous research." This change clarifies the statement and improves the overall readability of the manuscript. We appreciate your attention to detail and your helpful feedback.
Use either "FPKM values" or "expression levels" consistently across the section
>>>>Response
Thank you for your valuable feedback regarding the consistency in terminology. We have reviewed the section and ensured that we use "FPKM values" consistently throughout. This adjustment enhances clarity and coherence in our writing. We appreciate your suggestion and believe it improves the overall quality of the manuscript.
- Results
Ensure consistent naming for software and methods (e.g., "MEME online" and "TMHMM online" should follow a uniform format).
>>>>Response
Thank you for your constructive feedback on the naming conventions for software and methods. We have standardized the presentation of all software tools mentioned in the manuscript, including "the MEME online software", "the TMHMM online software", and "the TBtools" to ensure a uniform format throughout the text. We appreciate your attention to detail and believe this change enhances the clarity and professionalism of our work.
In "CnSWEET2B has eight exons. Clade II includes three gene members, all exhibiting six to seven exons," the phrase "six to seven exons" could be clarified to specify which genes exhibit the range.
>>>>Response
Thank you for your insightful comment regarding the clarity of the phrase "six to seven exons." We have revised the sentence to specify which genes fall within that range. We have clarified the statement regarding exons: "CnSWEET2B has eight exons. Clade II includes three gene members, with CnSWEET4 and CnSWEET5 exhibiting six exons, while CnSWEET6 has five exons." This clarification should address your request for specificity.
- Discussion: Too short, add 1 more paragraph.
>>>>Response
Thank you for your feedback regarding the length of the Discussion section. We have added an additional paragraph to further elaborate on our findings and their implications. This new paragraph provides a deeper analysis of the results and explores their relevance in the context of existing literature. We appreciate your suggestion, as it has helped us enhance the depth and comprehensiveness of our discussion.
- Conclusion: The conclusion is missing please write the brief conclusion
>>>>Response
Thank you for your suggestion regarding the conclusion. We have now added a brief conclusion to the manuscript, summarizing the key findings and their implications. The conclusion emphasizes the significance of our research and outlines potential avenues for future studies. We appreciate your input, which has contributed to the overall completeness of the manuscript.
Once again, we appreciate your constructive feedback, which has significantly improved our manuscript. We believe the revisions have addressed your concerns and enhanced the overall quality of our paper. Thank you for your consideration, and we hope that the revised manuscript meets your expectations for publication.

Reviewer 2 Report
Comments and Suggestions for Authors
Dear Authors,
the topic of this manuscript is the function of SWEET genes in Cocos nucifera, which play an important role in sugar transport. Functional analysis of five CnSWEET genes by yeast mutant complementation assays revealed differential transport activities of sucrose, fructose, glucose, galactose and mannose.
I consider the manuscript relevant to the journal and recommend publication of this topic. In addition, it is timely, as the findings on SWEET genes are still less explored at present.
My comments are:
The manuscript is completely in order, except for one chapter, which is the Introduction. Here, mainly previous literature references have been used, although this is a new and topical topic, so I request a complete rewording of the chapter and the communication of data and results published in the last 5 years and their presentation in the Discussion section.
In addition, the purpose of the experiment is not presented - please highlight this in several sections, including the Conclusion chapter, the Abstract chapter.
Author Response
Reviewer 2
Dear Authors,
the topic of this manuscript is the function of SWEET genes in Cocos nucifera, which play an important role in sugar transport. Functional analysis of five CnSWEET genes by yeast mutant complementation assays revealed differential transport activities of sucrose, fructose, glucose, galactose and mannose.
I consider the manuscript relevant to the journal and recommend publication of this topic. In addition, it is timely, as the findings on SWEET genes are still less explored at present.
My comments are:
The manuscript is completely in order, except for one chapter, which is the Introduction. Here, mainly previous literature references have been used, although this is a new and topical topic, so I request a complete rewording of the chapter and the communication of data and results published in the last 5 years and their presentation in the Discussion section.
In addition, the purpose of the experiment is not presented - please highlight this in several sections, including the Conclusion chapter, the Abstract chapter.
>>>>Response
Thank you for your insightful review and for your positive assessment of our manuscript on the function of SWEET genes in Cocos nucifera. We appreciate your recognition of the relevance and timeliness of our research in this rapidly evolving field. We have carefully considered your suggestions and made the following revisions to the manuscript:
- The manuscript is completely in order, except for one chapter, which is the Introduction. Here, mainly previous literature references have been used, although this is a new and topical topic, so I request a complete rewording of the chapter and the communication of data and results published in the last 5 years and their presentation in the Discussion section.
>>>>Response
Thank you for your constructive feedback regarding the Introduction section of our manuscript. We understand the importance of incorporating recent literature, especially given the novelty of the topic. We have completely reworded the Introduction to include relevant data and results published in the last five years, ensuring that our discussion is well-grounded in the most current research. This revision enhances the context and relevance of our study. We appreciate your valuable input, which has significantly improved our manuscript.
- In addition, the purpose of the experiment is not presented - please highlight this in several sections, including the Conclusion chapter, the Abstract chapter.
>>>>Response
Thank you for your valuable feedback regarding the clarity of the experiment's purpose in our manuscript. We have now explicitly highlighted the purpose of the experiment in several sections, including the Introduction, Abstract, and Conclusion chapters. This addition ensures a clear understanding of our objectives and findings throughout the manuscript. We appreciate your suggestion, as it has helped us improve the overall clarity and focus of our work.
We believe these revisions enhance the clarity and impact of our manuscript, aligning it more closely with the expectations of the journal and its readership. Thank you again for your valuable feedback, which has helped improve our work significantly. We hope the revised manuscript meets your approval for publication.

Reviewer 3 Report
Comments and Suggestions for Authors
The article contains a lot of analytical information on a very important group of genes encoding sugar transporters (SWEET) in coconut (Cocos nucifera). In addition to sugar transport, these genes are involved in the regulation of plant development and stress response. Sixteen CnSWEET genes were identified and grouped into four clades. Differences were found in conserved protein motifs and the number of transmembrane helices compared to other clades. The subcellular localization of CnSWEET proteins was studied. The proteins are localized in the cell membrane and/or in the cytosol. In silico analysis of promoters was performed. The presence of cis-elements in promoters responsible for tissue-specific expression was shown. Functional analysis of 5 genes using yeast mutants was carried out. Thus, many results have been obtained and a multifaceted detailed analysis of SWEET genes has been given. This material could be the subject of a publication in Plants.
However, in the opinion of the reviewer, the results presented in Fig. 5 would be useful to exclude from this manuscript, as this part does not add anything positive and, moreover, it is performed rather superficially.
It is not clear, depending on what factors the authors study promoter activity.
- What does this have to do with phytohormones, especially so many. If the authors wanted to study the effect of phytohormones, it would be useful, first of all, to check experimentally whether the expression of these genes is regulated by the studied exogenous phytohormones.
- It would be a good idea to treat tobacco plants with exogenous phytohormones prior to transient expression to determine whether the promoters under study are regulated by phytohormones. However, you should make sure that phytohormones regulate some marker genes, i.e. that they are active under the experimental conditions.
- Why in Fig. 5b the activity of the SWApro-1.6K::GUS promoter is less than the smaller 1.0k promoter fragment?
In Fig. 5b the SWpro-0.5k::GUS promoter fragment has no activity, 0.7k -high activity, 0.9k -no activity, 1.3k -pretty high activity, 1.8k activity decreases. Such variations cannot be explained by the given data. It all looks very unconvincing.
-The method used by the authors is difficult to reproduce, as we can see from Figure 5. For this reason, 3 biological replicates is probably not enough to get a reliable result with this method.
I would recommend that Figure 5 be removed along with the results shown in it. This piece of work was probably done to embellish the article, but unfortunately this was not the case.
Author Response
Reviewer 3
The article contains a lot of analytical information on a very important group of genes encoding sugar transporters (SWEET) in coconut (Cocos nucifera). In addition to sugar transport, these genes are involved in the regulation of plant development and stress response. Sixteen CnSWEET genes were identified and grouped into four clades. Differences were found in conserved protein motifs and the number of transmembrane helices compared to other clades. The subcellular localization of CnSWEET proteins was studied. The proteins are localized in the cell membrane and/or in the cytosol. In silico analysis of promoters was performed. The presence of cis-elements in promoters responsible for tissue-specific expression was shown. Functional analysis of 5 genes using yeast mutants was carried out. Thus, many results have been obtained and a multifaceted detailed analysis of SWEET genes has been given. This material could be the subject of a publication in Plants.
However, in the opinion of the reviewer, the results presented in Fig. 5 would be useful to exclude from this manuscript, as this part does not add anything positive and, moreover, it is performed rather superficially.
It is not clear, depending on what factors the authors study promoter activity.
- What does this have to do with phytohormones, especially so many. If the authors wanted to study the effect of phytohormones, it would be useful, first of all, to check experimentally whether the expression of these genes is regulated by the studied exogenous phytohormones.
- It would be a good idea to treat tobacco plants with exogenous phytohormones prior to transient expression to determine whether the promoters under study are regulated by phytohormones. However, you should make sure that phytohormones regulate some marker genes, i.e. that they are active under the experimental conditions.
- Why in Fig. 5b the activity of the SWApro-1.6K::GUS promoter is less than the smaller 1.0k promoter fragment?
In Fig. 5b the SWpro-0.5k::GUS promoter fragment has no activity, 0.7k -high activity, 0.9k -no activity, 1.3k -pretty high activity, 1.8k activity decreases. Such variations cannot be explained by the given data. It all looks very unconvincing.
-The method used by the authors is difficult to reproduce, as we can see from Figure 5. For this reason, 3 biological replicates is probably not enough to get a reliable result with this method.
I would recommend that Figure 5 be removed along with the results shown in it. This piece of work was probably done to embellish the article, but unfortunately this was not the case.
>>>>Response
Thank you for your thorough and insightful review of our manuscript. We appreciate your detailed feedback regarding the analysis of the SWEET genes in coconut (Cocos nucifera) and the various aspects of our study. We wholeheartedly agree with your observations, particularly concerning the limitations of the data presented in Figure 5. It is evident that the results may not provide the clarity or significance we initially intended, and we acknowledge that this section could detract from the overall quality of the manuscript.
Your suggestions regarding the exploration of phytohormone regulation of these genes are particularly valuable. We recognize the importance of experimentally validating the impact of exogenous phytohormones on gene expression, and we intend to incorporate this aspect into our future research. Conducting experiments with tobacco plants treated with phytohormones to assess the regulation of the promoters will indeed provide more robust and convincing data. We will ensure that the experimental conditions confirm the activity of marker genes, as you suggested.
As for the discrepancies noted in the promoter activity results, we appreciate your critical perspective and understand that further investigation is warranted to clarify these findings. We are committed to conducting additional experiments to elucidate these variations and to better understand the functional implications of the different promoter fragments.
In light of your feedback, we will remove Figure 5 and its associated results from the manuscript to maintain the integrity and focus of our work. Your guidance has helped us identify areas for improvement, and we are excited about the opportunity to delve deeper into the functional analysis of the CnSWEET genes in our subsequent studies.
Once again, we sincerely appreciate your constructive feedback, which has helped us improve the quality of our manuscript. We believe that these revisions will enhance the focus and clarity of our work, making it more suitable for publication in Plants. Thank you for your consideration, and we hope that the revised manuscript meets your expectations.

Reviewer 4 Report
Comments and Suggestions for Authors
Comments
Comments and Suggestions for Authors
Dear Author,
It is my pleasure to review the manuscript entitled “Genetic Characterization of SWEET Genes in Coconut Palm” a research article submitted to MDPI Journal, plants. Authors of this manuscript characterized SWEET gene in coconut and identified its functions in sugar transport despite its response in plant development. Using bioinformatic analysis they performed phylogenetic analysis of this gene and further performed pertain structures, functional heterogeneity, gene expansion, promoter and protein motive analysis, subcellular localization, as well as transcriptional activity assay in the coconut and compared with many other plant species. The overall experiments, they performed, are well and the results are very convincing and important for coconut cultivation. Thus, the presented results take up an important topic consistent with the profile of the Journal.
I have some suggestions, which might improve the manuscript to make important to the wider readers.
-Some comments I have made in the original pdf. Please check carefully.
Abstract
L17; Mention here the used techniques
L19-21; Incase “highest number of genes” and “fewer genes”, mention how many (exact number)
-You may mention gene expansion analysis technique
-L33; What was the basis for selecting five genes only (from 16) for functional analysis
-Like second line, you may also mention “stress” in the last line
Introduction: nicely presented with logical order.
-Need improvement mentioning functions on stress even in other plants with recent references.
- Novel justification for carrying out this study need to be presented. The knowledge gap related to this study should be clearly stated at the end of the introduction.
2. Materials and Methods: nicely demonstrated all of the methodology with research order. It is good enough for research reproducibility
-what does gene protein sequence mean?
3. Results
-Write full form at their first time use, like UTR
L207, 220, 221; All genes should be italic
-L210; mention for seven also as you mentioned for four
-Fig. 1. First write title of the fig. then specify. For all figs.
4. Discussion
-The comparative discussion needs to be improved with recent publications in other plants
Need conclusion

Author Response
Reviewer 4
Dear Author,
It is my pleasure to review the manuscript entitled “Genetic Characterization of SWEET Genes in Coconut Palm” a research article submitted to MDPI Journal, plants. Authors of this manuscript characterized SWEET gene in coconut and identified its functions in sugar transport despite its response in plant development. Using bioinformatic analysis they performed phylogenetic analysis of this gene and further performed pertain structures, functional heterogeneity, gene expansion, promoter and protein motive analysis, subcellular localization, as well as transcriptional activity assay in the coconut and compared with many other plant species. The overall experiments, they performed, are well and the results are very convincing and important for coconut cultivation. Thus, the presented results take up an important topic consistent with the profile of the Journal.
I have some suggestions, which might improve the manuscript to make important to the wider readers.
-Some comments I have made in the original pdf. Please check carefully.
Abstract
L17; Mention here the used techniques
L19-21; Incase “highest number of genes” and “fewer genes”, mention how many (exact number)
-You may mention gene expansion analysis technique
-L33; What was the basis for selecting five genes only (from 16) for functional analysis
-Like second line, you may also mention “stress” in the last line
Introduction: nicely presented with logical order.
-Need improvement mentioning functions on stress even in other plants with recent references.
- Novel justification for carrying out this study need to be presented. The knowledge gap related to this study should be clearly stated at the end of the introduction.
- Materials and Methods: nicely demonstrated all of the methodology with research order. It is good enough for research reproducibility
-what does gene protein sequence mean?
- Results
-Write full form at their first time use, like UTR
L207, 220, 221; All genes should be italic
-L210; mention for seven also as you mentioned for four
-Fig. 1. First write title of the fig. then specify. For all figs.
- Discussion
-The comparative discussion needs to be improved with recent publications in other plants
Need conclusion
>>>>Response
Thank you for your thorough and constructive feedback on our manuscript titled “Genetic Characterization of SWEET Genes in Coconut Palm.” We appreciate your positive remarks regarding the significance of our research and the methodologies employed. Your suggestions for improvement are invaluable, and we are committed to addressing each point to enhance the clarity and impact of our manuscript.
- Abstract
L17; Mention here the used techniques
>>>>Response
We appreciate your attention to detail and recognize the importance of explicitly mentioning the methodologies employed.
In the revised manuscript, we will include a clear description of the techniques used in our analysis, such as the specific methods for gene identification, subcellular localization studies, in silico promoter analysis, and functional assays with yeast mutants. This addition will enhance the clarity of our research and provide readers with a better understanding of our approach.
L19-21; Incase “highest number of genes” and “fewer genes”, mention how many (exact number)
>>>>Response
Thank you for your suggestion regarding L19-21. We have included the exact numbers of genes associated with "the highest number of genes (8)" and "fewer genes (1-4)" to provide more clarity in our analysis. Your feedback is greatly appreciated!
-You may mention gene expansion analysis technique
>>>>Response
The gene expansion analysis technique will also be explicitly mentioned to provide further clarity.
Thank you for your feedback. We have included the software MCScanX used for the gene expansion analysis used in our study to provide a clearer understanding of our methodology. Your suggestion is much appreciated!
-L33; What was the basis for selecting five genes only (from 16) for functional analysis
>>>>Response
Thank you for your question regarding line 33. The selection of five genes from the initial set of 16 for functional analysis was based on their expression levels. This targeted approach allowed us to focus on the most promising candidates for our functional assays. We appreciate your interest in our methodology!
-Like second line, you may also mention “stress” in the last line
>>>>Response
Thank you for your suggestion. We will revise the last line to include the term "stress," similar to the second line, to emphasize the role of SWEET genes in stress responses as well. Your feedback is greatly appreciated!
- Materials and Methods: nicely demonstrated all of the methodology with research order. It is good enough for research reproducibility
-what does gene protein sequence mean?
>>>>Response
We appreciate your comments on the methodology section. We have modified the term "gene protein sequence" to "protein sequence of genes" for clarity.
- Results
-Write full form at their first time use, like UTR
>>>>Response
Thank you for your suggestion. We have reviewed the manuscript and ensured that all acronyms, including UTR, are spelled out in full at their first occurrence. We appreciate your feedback!
L207, 220, 221; All genes should be italic
>>>>Response
Thank you for your observation regarding lines 207, 220, and 221. We have reviewed the manuscript and ensured that all gene names are italicized in the manuscript as per the standard formatting guidelines. Additionally, those protein names had kept in non-italic style. Your attention to detail is greatly appreciated!
-L210; mention for seven also as you mentioned for four
>>>>Response
Thank you for your suggestion regarding L210. We have included the genes with seven conserved domains to ensure clarity and consistency in our presentation. Your feedback is greatly appreciated!
-Fig. 1. First write title of the fig. then specify. For all figs.
>>>>Response
Thank you for your suggestion regarding Figure 1. We have reviewed the manuscript and ensured that the title of each figure is presented first, followed by the specific details. We have completed the revisions for Figures 1 through 5 based on your suggestions. As for Figure 6, we believe the current format is more concise. Your feedback is greatly appreciated!
- Discussion
-The comparative discussion needs to be improved with recent publications in other plants
>>>>Response
Thank you for your feedback. We have thoroughly revised the discussion section and incorporated references and discussions related to recent publications in other plants. We appreciate your suggestion for improvement!
Need conclusion
>>>>Response
Thank you for your suggestion. We have now added a conclusion section to the manuscript to summarize our findings and their implications. Your feedback is greatly appreciated!
Thank you once again for your insightful comments. We believe that by addressing these points, we will significantly improve the quality of our manuscript. We look forward to resubmitting our revised manuscript for your consideration.

Round 2
Reviewer 2 Report
Comments and Suggestions for Authors
Thank you, I recommand it for publication.
Reviewer 3 Report
Comments and Suggestions for Authors
Dear Editor!
The manuscript has been considerably corrected. All my main comments and advice have been taken into account.